# *Pseudomonas donghuensis* HYS *gtrA*/*B*/*II* Gene Cluster Contributes to Its Pathogenicity toward *Caenorhabditis elegans*

**DOI:** 10.3390/ijms221910741

**Published:** 2021-10-04

**Authors:** Yaqian Xiao, Panning Wang, Xuesi Zhu, Zhixiong Xie

**Affiliations:** Hubei Key Laboratory of Cell Homeostasis, College of Life Sciences, Wuhan University, Wuhan 430072, China; 2018102040012@whu.edu.cn (Y.X.); 2015202040019@whu.edu.cn (P.W.); 2019202040032@whu.edu.cn (X.Z.)

**Keywords:** *Pseudomonas donghuensis* HYS, GtrA, GtrB, GtrII, virulence, *Caenorhabditis elegans*

## Abstract

*Pseudomonas donghuensis* HYS is more virulent than *P. aeruginosa* toward *Caenorhabditis elegans* but the mechanism underlying virulence is unclear. This study is the first to report that the specific gene cluster *gtrA/B*/*II* in *P. donghuensis* HYS is involved in the virulence of this strain toward *C. elegans*, and there are no reports of GtrA, GtrB and GtrII in any *Pseudomonas* species. The pathogenicity of *P. donghuensis* HYS was evaluated using *C. elegans* as a host. Based on the prediction of virulence factors and comparative genomic analysis of *P. donghuensis* HYS, we identified 42 specific virulence genes in *P. donghuensis* HYS. Slow-killing assays of these genes showed that the *gtrAB* mutation had the greatest effect on the virulence of *P. donghuensis* HYS, and GtrA, GtrB and GtrII all positively affected *P. donghuensis* HYS virulence. Two critical GtrII residues (Glu^47^ and Lys^480^) were identified in *P. donghuensis* HYS. Transmission electron microscopy (TEM) showed that GtrA, GtrB and GtrII were involved in the glucosylation of lipopolysaccharide (LPS) O-antigen in *P. donghuensis* HYS. Furthermore, colony-forming unit (CFU) assays showed that GtrA, GtrB and GtrII significantly enhanced *P. donghuensis* HYS colonization in the gut of *C. elegans*, and glucosylation of LPS O-antigen and colonization in the host intestine contributed to the pathogenicity of *P. donghuensis* HYS. In addition, experiments using the worm mutants ZD101, KU4 and KU25 revealed a correlation between *P. donghuensis* HYS virulence and the TIR-1/SEK-1/PMK-1 pathways of the innate immune p38 MAPK pathway in *C. elegans*. In conclusion, these results reveal that the specific virulence gene cluster *gtrA*/*B*/*II* contributes to the unique pathogenicity of HYS compared with other pathogenic *Pseudomonas*, and that this process also involves *C. elegans* innate immunity. These findings significantly increase the available information about GtrA/GtrB/GtrII-based virulence mechanisms in the genus *Pseudomonas*.

## 1. Introduction

Virulence is a pathogen’s ability to cause harm or disease, involving nutrient competition and requiring self-protection, and can thereby protect them from predators and hostile environments [1,2]. When invading a host, pathogens need to respond rapidly to adverse situations, including by activating associated virulence-related programs and escaping the host immune defenses.

*Pseudomonas*, a genus of obligate aerobic Gram-negative bacilli, belongs to the class of Gammaproteobacteria [3,4,5,6]. This genus contains many pathogen species, including *P. aeruginosa*, *P. putida*, *P. fluorescens* and *P. syringae* [7,8,9]. *P. donghuensis* exhibits antifungal and plant-promoting activities [10,11]. *P. donghuensis* HYS exhibits a high iron-chelating capacity and secretes pyoverdine and a large quantity of 7-hydroxytropolone, a nonfluorescent substance that exerts antibacterial and antifungal effects [12,13,14]. *P. donghuensis* HYS is the type strain of this new species and can thus be used as a representative of the whole species when conducting pathogenic research. In a previous study, this strain was shown more virulent to *Caenorhabditis elegans* than human opportunistic pathogens *P. aeruginosa* PAO1 or PA14 under the same culture conditions, suggesting the possible existence of a new pathogenic mechanism [15]. However, the pathogenic mechanism of *P. donghuensis* HYS remains unclear.

*C. elegans* is used as a model organism for the analysis of pathogenic bacteria [16], as its small body size, simple body structure, short lifespan and completely sequenced genome [17,18,19,20], and this worm can interact with many known human pathogens and has a functional innate immune response [21]. *C. elegans* provides us with a model for studying virulence factors, pathogenetic mechanisms and host-pathogen interactions [22,23]. In our previous work, *P. donghuensis* HYS caused a high level of mortality in *C. elegans* in a slow-killing experiment [13,15,16].

Lipopolysaccharide (LPS) is an important pathogenic factor composed of three regions: the lipid A membrane anchor, an inner core and repeating O-antigen subunits [24]. In *Shigella flexneri*, O-antigen glucosylation is mediated by temperate bacteriophages, which encode a three-gene cluster that contains *gtrA*, *gtrB* and *gtr _[type]_*. GtrA family members are often involved in the synthesis of cell-surface polysaccharides [25], GtrB is polyisoprenyl phosphate glycosyltransferase, and Gtr_[type]_ encodes a species-specific glucosyltransferase [26], which specifically attaches a glucosyl residue to the appropriate rhamnose of the O-antigen chain [27,28,29]. However, *gtrA*, *gtrB* and *gtrII* have not been reported in *Pseudomonas*.

In this study, we first used the Virulence Factor of Pathogenic Bacteria (VFDB) database to predict virulence-related genes in *P. donghuensis* HYS. We then screened for specific virulence genes in *P. donghuensis* HYS through comparative genomic analysis of whole genome sequences of *P. donghuensis* HYS and six common pathogenic *Pseudomonas* model strains. We thereby obtained 42 specific virulence genes in *P. donghuensis* HYS and found that the *gtrA*/*B*/*II* gene cluster does not exist in all pathogenic *Pseudomonas*. Our work further revealed that the *gtrA*/*B*/*gtrII* gene cluster positively affected virulence. In addition, we tested the correlation between *P. donghuensis* HYS virulence and *C. elegans* innate immunity. Our data suggest that the *gtrA*, *gtrB* and *gtrII* gene cluster plays a role in the virulence of *P. donghuensis* HYS, and this result may facilitate increased comprehension of the pathogenesis process and mechanisms of *P. donghuensis,* as well as of the genus *Pseudomonas* more generally.

## 2. Results

### 2.1. Screening of Specific Virulence Genes Based on Virulence Factor Prediction and Comparative Genomic Analysis

In a previous study, we found that *C. elegans* fed with *P. donghuensis* HYS had a significantly shorter lifespan than those fed with *P. aeruginosa* [15], which indicates that the strain is highly virulence and potentially employs a different virulence mechanism. We first predicted virulence factors of *P. donghuensis* HYS using the VFDB database. The prediction criteria were a protein sequence identity of more than 40% and a difference in length between protein sequences of less than 30%. As a result, we obtained 440 genes that may be related to virulence in *P. donghuensis* HYS. Then, we performed comparative genomic analysis of whole genome sequences of the *P. donghuensis* HYS strain and six model strains of *Pseudomonas*, which were *P. aeruginosa* PAO1, *P. aeruginosa* PA14, *P. putida* KT2440, *P. syringae* B728a, *P. syringae* pv *tomato* DC3000 and *P. fluorescens* Pfo-1. The clustering criteria were a protein sequence identity of more than 50% and a difference in length between protein sequences of less than 30%. Cluster analysis of genes and gene families revealed 1022 specific genes and 10 gene families of *P. donghuensis* HYS (Figure 1A,B). Among the 1022 genes, 584 specific genes were annotated by the Clusters of Orthologous Groups (COG) database. The functional classifications of specific annotated genes were mainly cellular (135 unigenes), information (105 unigenes), metabolism (251 unigenes) and poorly characterized (93 unigenes) (Figure 1C). The 440 virulence genes predicted by the VFDB database were compared with the 584 specific genes obtained by comparative genomic analysis, we screened 42 of 440 virulence genes that are also specific genes of *P. donghuensis* HYS. Therefore, these 42 genes are specific virulence genes in *P. donghuensis* HYS (Appendix A).

The virulence of *P. donghuensis* HYS is stronger than that of other *Pseudomonas* species, which might be related to it having these specific virulence genes. We classified and knocked out 42 specific virulence genes and performed slow-killing assays. *E. coli* OP50, a traditional food source for *C. elegans*, was used as a negative control, and wild-type HYS was used as a positive control. The survival data show that the average LT_50_ value (the time required to kill 50% of worms exposed to *E. coli* OP50) was 12.38 ± 0.36 days (*n* = 3). However, worms exposed to *P. donghuensis* HYS died quickly (the LT_50_ value decreased to 3.36 ± 0.46 days; *n* = 3). By comparison, the knockout strains Δ*hemN/N*, Δ*hurR/hmuVUTS*, Δ*gtrAB* and Δ*irgA* had significantly reduced virulence (the LT_50_ values increased to 5.54 ± 0.22 days, 4.91 ± 0.16 days, 6.33 ± 0.27 days and 5.24 ± 0.19 days, respectively; *n* = 3) (Figure 1D), while other knockout strains showed no significant reductions in virulence (Appendix A). The double-knockout strain Δ*gtrAB* had the most significant reduction in virulence (the LT_50_ value was 6.33 ± 0.27 days; *n* = 3), nematodes fed Δ*gtrAB* survived for 3.0 days longer than those fed wild-type HYS (the LT_50_ value was 3.36 ± 0.46 days; *n* = 3). Interestingly, there are no reports of GtrA and GtrB in *Pseudomonas*. In addition, *gtrII* is behind *gtrB* in the *P. donghuensis* HYS genome also found to be unique to *P. donghuensis* HYS through comparative genome analysis. However, the function of GtrII is not clear, and GtrII has not been annotated in VFDB, KEGG (Kyoto Encyclopedia of Genes and Genomes), COG or GO (Gene Ontology) databases through analyses genome sequencing data; therefore, GtrII was not screened through VFDB prediction and COG database annotation. *S. flexneri* has a three-gene cluster containing *gtrA*, *gtrB* and *gtr*_[type]_; *gtr*_[type]_ encodes a species-specific glucosyltransferase, which specifically attaches a glucosyl residue to the appropriate rhamnose of the O-antigen chain. The *gtrA*, *gtrB* and *gtrII* genes may be involved in the virulence of HYS and may provide this strain with special virulence characteristics compared with pathogenic *Pseudomonas*.

### 2.2. The Particularity of GtrA, GtrB and GtrII in Pseudomonas

In order to obtain clues about the evolution of the gene combination, we compared the protein homology of GtrA, GtrB and GtrII using BLASTP and set the threshold at 40%. The comparison results showed that most *Pseudomonas*, including those that are pathogenic, contain one or two genes of the *gtrA*/*B*/*II* gene cluster, and the arrangement of GtrA/B/II only exists in four strains other than HYS (Figure 2A). Five strains, including *P. donghuensis* HYS, were isolated from different biological samples in different countries, and there is evidence of geographical isolation (Table 1). To better understand the evolutionary position of GtrA, GtrB and GtrII in *P. donghuensis* HYS, we constructed phylogenetic trees according to the amino acid sequences of GtrA, GtrB and GtrII in bacteria using the neighbor-joining method with a bootstrap value of 1000 (Figure 2B). The phylogenetic trees show that the GtrA/B/II cluster in *P. donghuensis* HYS is grouped most closely with that in *P. vranovensis*, followed by *Pseudomonas* sp. WS 5059. These results suggest that the *gtrA*/*B*/*II* gene cluster has an evolutionary particularity, which is that the gene combination did not form by gradual evolution but formed in a short time in a changeable environment to meet the needs of bacterial survival and growth.

### 2.3. The Virulence of P. donghuensis HYS Is Optimized by the gtrA/B/II Virulence Gene Cluster in a C. elegans Slow-Killing Assay

*P. donghuensis* HYS is highly virulent toward *C. elegans*. We performed a slow-killing assay to test bacterial virulence against *C. elegans*. To test the functions of *gtrA*, *gtrB* and *gtrII* in virulence, we constructed single-knockout strains Δ*gtrA*, Δ*gtrB* and Δ*gtrII*, the double-knockout strain Δ*gtrAB* and the triple-knockout strain Δ*gtrABII* and performed a slow-killing assay utilizing *C. elegans*. Survival analysis showed that the lifespan and LT_50_ value of *C. elegans* fed the single-knockout strains were approximately 1.5 times greater than those fed the wild-type strain (the LT_50_ values increased to 4.39 ± 0.13 days, 4.77 ± 0.16 days and 5.09 ± 0.19 days, respectively; *n* = 3); those fed the double-knockout strain Δ*gtrAB* and the triple-knockout strain Δ*gtrABII* survived for approximately twice as long as those fed the wild-type strain (the LT_50_ values increased to 6.33 ± 0.27 days and 6.02 ± 0.23 days, respectively; *n* = 3) (Figure 3A).

We performed single-gene complementation in the single-knockout strains and conducted a slow-killing assay. Restoration of virulence in the complemented strains further confirmed the functions of these three genes in bacterial virulence (the LT_50_ values decreased from 4.04 ± 0.13, 4.23 ± 0.14 and 4.41 ± 0.10 days to 3.01 ± 0.10, 3.32 ± 0.12 and 3.58 ± 0.08 days, respectively; *n* = 3) (Figure 3B). In addition, we also performed single-gene complementation and three-gene complementation in the triple-knockout strain Δ*gtrABII*, and the complemented strains were assayed in the slow-killing experiment. Survival analysis showed that single-gene complementation in the triple-knockout strain Δ*gtrABII* resulted in the slight restoration of virulence but could not return it to the level of the wild-type strain. Three-gene complementation in the triple-knockout strain Δ*gtrABII* could restore virulence to the wild-type level (the LT_50_ values decreased from 5.50 ± 0.19 to 3.61 ± 0.10 days; *n* = 3) (Figure 3C). These results further confirm that *gtrA*, *gtrB* and *gtrII* form a virulence gene cluster. In addition, we tested the growth of the mutants, and it was found that the growth trend of the knockout strains was basically the same as that of the wild-type strain, without obvious growth defects (Figure 3D,E). Thus, the reduced virulence was not due to a deficiency in bacterial growth, which demonstrated that the reduced virulence of the knockout strains was indeed caused by the deletion of gene functions. These results suggest that the *gtrA*/*B*/*II* gene cluster plays a crucial role in the virulence of *P. donghuensis* HYS.

### 2.4. Identification of Critical GtrII Residues in P. donghuensis HYS

To further understand the virulence effect of GtrII in *P. donghuensis* HYS, the GtrII protein sequence of *P. donghuensis* HYS was compared with BLASTP in NCBI, which showed that GtrII belonged to the GtrII protein superfamily and was a transmembrane protein with nine transmembrane regions (Appendix A). In *S. flexneri*, three critical residues (Glu ^40^, Phe ^414^, and Lys ^478^) are conserved in Gtr_[type]_ [34]. However, Bioinformatics was limited in finding conserved residues between them as a result of the low sequence homology and different protein lengths among Gtr_[type]_. Based on the conservative localization of critical residues among Gtr_[type]_, GtrII residues (Glu^47^, Phe^430^, Phe^431^ and Lys^480^) were manually identified and selected for point mutating to alanine (Appendix A). Accordingly, we constructed strains with E47A, F430A, F431A and K480A point mutations in the genome and performed a slow-killing assay to test the virulence of the mutants against *C. elegans*. The results show that the virulence function of GtrII was abolished in the E47A and K480A mutants (the LT_50_ values increased from 3.36 ± 0.46 days to 5.01 ± 0.17 days and 5.05 ± 0.14 days, respectively; *n* = 3) (Figure 4A), which is consistent with Δ*gtrII* (the LT_50_ value was 5.09 ± 0.19 days; *n* = 3). F430A and F431A mutants did not show the same reduced virulence as Δ*gtrII*. In order to exclude the possible influence of the growth of the point mutation strains on virulence, we tested the growth of the mutants. The growth trend of the point mutation strains was found to be basically the same as that of the wild-type strain, without obvious growth defects (Figure 4B). A colony-forming unit (CFU) assay was performed to determine the colonization of point mutation strains in the gut of *C. elegans* (Figure 4C). E47A and K480A mutations were observed to significantly inhibit the number of living bacteria colonized in the gut of *C. elegans*. However, we observed that F430A and F431A mutations did not inhibit *P. donghuensis* HYS colonization in *C. elegans*. These results show that two critical GtrII residues (Glu47 and Lys480) control the virulence function of GtrII in *P. donghuensis* HYS. At present, there are no reports describing the role of GtrII in bacterial virulence.

### 2.5. Surface Topology of Strains Revealed by Transmission Electron Microscopy

To test the correlation between *gtrA*, *gtrB* and *gtrII* and the glucosylation of LPS O-antigen, bacteria were visualized by transmission electron microscopy (TEM) (Figure 5). Wild-type HYS had dense surface material extending about 20 nm beyond the outer membrane (between the two arrowheads). By contrast, the exteriors of Δ*gtrA*, Δ*gtrB*, Δ*gtrII*, Δ*gtrAB* and Δ*gtrABII* were more diffuse. The filamentous material of Δ*gtrA* extended around 34 nm from the outer membrane, and the filamentous material of Δ*gtrB*, Δ*gtrII*, Δ*gtrAB* and Δ*gtrABII* extended around 40 nm from the outer membrane. Δ*gtrABII*/pABII had dense surface material extending around 20 nm from the outer membrane, which was basically the same as that of the wild-type strain. The glucosylation of LPS O-antigen in *P. donghuensis* HYS dramatically shortens the O antigen, halving the distance that it extends beyond the outer membrane. GtrA, GtrB and GtrII may be involved in the glucosylation of LPS O-antigen in *P. donghuensis* HYS.

### 2.6. Estimation of P. donghuensis HYS CFU within the C. elegans Gut

*C. elegans* was exposed to *P. donghuensis* HYS in the same conditions as those of the slow-killing assay. We measured CFU to assess the colonization of deletion mutants in the gut of *C. elegans* (Figure 6). To rule out the possibility of an effect of different pharyngeal pumping rates on bacterial load, we quantitatively measured pathogen-clearance ability by first ensuring that N2 worms were inoculated with a similar number of bacteria. We observed that deletion mutants of *gtrB*, *gtrII*, *gtrAB* and *gtrABII* significantly suppressed *P. donghuensis* HYS colonization in the gut of nematodes. The deletion mutant of *gtrA* also significantly inhibited the intestinal colonization of *P. donghuensis* HYS, but the effect was weaker than that of the other mutants. Colonization in the host is the first and key step of exerting pathogenicity for a bacterium. These results suggest that the involvement of GtrA, GtrB and GtrII in bacterial virulence may be related to the colonization of bacteria in the host gut.

### 2.7. GtrA/B/II-Involved Virulence Is Related to C. elegans Innate Immunity

Unlike higher organisms, *C. elegans* lacks adaptive immune pathways, and only innate immune pathways play an important role in resisting pathogens, oxidative stress and other types of stress [21]. To investigate the worm response to *P. donghuensis* HYS virulence, we utilized relevant mutants to conduct a slow-killing assay. The mutant worms ZD101, KU4 and KU25 are deficient in the Tir-1, SEK-1 and PMK-1 pathways of the p38 MAPK pathway, respectively. After feeding on *P. donghuensis* HYS, mutant *C. elegans* worms were more sensitive and died more quickly than N2 worms. However, when we used the *gtrAB* deletion mutant as an alternative food source, their viability was significantly improved, illustrating that the strain Δ*gtrAB* had lower virulence than the wild-type strain when tested as a food source for mutant worms. The LT_50_ values of mutant worms ZD101 (Figure 7A), KU4 (Figure 7B) and KU25 (Figure 7C) fed *P. donghuensis* HYS were approximately 49.25%, 64.78% and 33.13% lower than that of N2 worms fed *P. donghuensis* HYS (the LT_50_ value decreased from 3.35 ± 0.07 to 1.69 ± 0.04, 1.18 ± 0.03 and 2.24 ± 0.05 days, respectively; *n* = 3). In addition, when the Δ*gtrAB* deletion mutant was provided as an alternative food source, the LT_50_ values of mutant worms decreased by approximately 54.66%, 58.17% and 49.52% compared with that of N2 worms (the LT_50_ value decreased from 6.24 ± 0.22 days to 2.82 ± 0.05 days, 2.61 ± 0.08 days and 3.15 ± 0.10 days, respectively; *n* = 3). These percentages suggest a correlation between TIR-1/SEK-1/PMK-1 of the innate immunity p38 MAPK pathway in *C. elegans* and the *gtrAB*-related virulence of *P. donghuensis* HYS.

## 3. Discussion

The virulence of *P. donghuensis* HYS is stronger than that of pathogenic *P. aeruginosa* PAO1 or *P. aeruginosa* PA14 [15]. Based on bioinformatics analysis and functional identification, we identified *gtrA*/*B*/*II* as a specific virulence gene cluster in *P. donghuensis* HYS compared with other pathogenic *Pseudomonas*, being found to exists in only four strains other than *P. donghuensis* HYS. A slow-killing assay confirmed that *gtrA*, *gtrB* and *gtrII* form a virulence gene cluster, and only in the presence of all three genes will it recover its virulence effects. GtrA family members are often involved in the synthesis of cell-surface polysaccharides [25], and GtrB is polyisoprenyl phosphate glycosyltransferase.Gtr_[type]_ encodes a species-specific glucosyltransferase [26], which specifically attaches a glucosyl residue to the appropriate rhamnose of the O-antigen chain [27,28]. At present, the virulence function of GtrII is unclear, and it has not been annotated in many databases. The GtrII protein belongs to the GtrII protein superfamily and is a transmembrane protein with nine transmembrane regions. Currently, Gtr_[type]_ has only been reported in *S. flexneri* and is involved in the modification of the O-antigen. However, bioinformatics was of limited use in finding conserved residues between them as a result of the low sequence homology and different protein lengths among Gtr_[type]_. To further understand the function of GtrII, the residues Glu^47^, Phe^430^, Phe^431^ and Lys^480^ were manually selected for E47A, F430A, F431A and K480A mutagenesis, respectively. The E47A and K480A mutations were found to abolish the virulence function of GtrII in *P. donghuensis* HYS, while the F430A and F431A mutations did not completely eliminate the virulence function of GtrII. We found that changing critical residues (Glu^47^ and Lys^480^) did not affect the structure of the GtrII protein. However, changing critical residues affected the colonization of the mutants and thus affected their virulence.

In *S. flexneri* 5a, *gtrA*, *gtrB* and *gtrV* are involved in the glucosylation of LPS O-antigen [29,34,35]. Glucosylation of LPS O-antigen induces a transition from a linear to helical comformation with the glucosyl residue exposed to the outside of the helix, forming a more compact structure than unglucosylated LPS [36]. This dramatically shortens the O antigen, halving the distance that it extends beyond the outer membrane. In the LPS model, the O antigen is only 11 nm under the action of glucosylation, while in the absence of glucosylation, the O antigen would extend about 21 nm beyond the outer membrane. In *S. flexneri*, the exterior of M90T is a dense material extending around 35 nm from the outer membrane. In contrast, M90TΔ*gtr* has more diffuse material extending about 70 nm beyond the outer membrane [35]. In *P. donghuensis* HYS, wild-type HYS has dense surface material extending about 20 nm beyond the outer membrane. By contrast, the exteriors of mutants are more diffuse material extending around 40 nm from the outer membrane. In addition, the surface of Δ*gtrABII*/pABII was basically the same as that of the wild-type HYS, filamentous material that extended around 20 nm beyond the outer membrane. Glucosylation of LPS O antigen dramatically shortening of the O antigen, also halving the distance that it extends beyond the outer membrane. These results are consistent with the results of glycosylation of *S. flexneri* O-antigen [36,37], indicating that GtrA, GtrB and GtrII may be involved in the glycosylation of LPS O-antigen. These results are consistent with the toxicity test results, indicating that the involvement of GtrA, GtrB and GtrII in bacterial toxicity may be related to the glycosylation of LPS O-antigen.

Colonization in the host is the first and key step of exerting pathogenicity for a bacterium [38], we show that the colonization of the knockout strains of *gtrA*, *gtrB* and *gtrII* in the gut of *C*. *elegans* was significantly reduced compared with that of the wild type. The *gtrA*/*B*/*II* gene cluster optimizes the virulence of *P. donghuensis* HYS toward *C. elegans* through colonization increased in the gut of *C*. *elegans*. At present, there are no reports on the effect of GtrA, GtrB and GtrII on bacterial virulence resulting from changes in bacterial colonization in the host intestine.

Unlike higher organisms, *C. elegans* only has innate immune pathways, which play an important role in resisting pathogens and oxidative stress [39,40,41]. The MAPK, TGF-β and DAF-2/DAF-16 signaling pathways are conserved innate immune signaling pathways necessary for *C. elegans* to resist pathogens. The p38 MAPK pathway belongs to the MAPK pathway, which is the most crucial pathway in intestinal innate immunity, making these pathways suitable for exploring the correlation between pathogenicity and host response [37]. TIR-1 (MAPKKK), SEK-1 (MAPKK) and PMK-1 (MAPK) pathways of the p38 MAPK pathway are critical in responding to infection [42]. In this study, we explored the defense response of *C. elegans* to a virulence infection with *P. donghuensis* HYS. We utilized the relevant mutant worms to conduct a slow-killing assay. Mutant ZD101 is deficient in the Toll/I-1 receptor (TIR-1) pathway, mutant KU4 is deficient in the SEK-1 pathway and mutant KU25 is deficient in the PMK-1 pathway. Mutant *C. elegans* died shortly after feeding on *P. donghuensis* HYS, but their viability was significantly improved when we used the *gtrAB* deletion mutant as an alternative food source. The results suggest a response to the *gtrA*/*B*/*II*-optimized virulence of *P. donghuensis* HYS that is mediated by the TIR-1, SEK-1 and PMK-1 components of the innate immune p38 MAPK pathway, which has not been previously reported. These pathways may supplement the virulence mechanism based on the *gtrA*/*B*/*II* gene cluster.

In summary, we found that the specific virulence gene cluster *gtrA*/*B*/*II* optimizes the virulence of *P. donghuensis* HYS toward *C. elegans* through colonization increased in the gut of *C*. *elegans*, and this process also involves *C. elegans* innate immunity. At present, there are no reports on the effect of GtrA, GtrB and GtrII on bacterial virulence resulting from changes in bacterial colonization in the host intestine, which suggests a novel virulence mechanism that differs from those used by other pathogenic *Pseudomonas*. Therefore, the *P. donghuensis* virulence phenotype differs from that of *P. aeruginosa*. These findings significantly expand upon the available information regarding *gtrA*/*B*/*II*-based virulence mechanisms in the genus *Pseudomonas*.

## 4. Materials and Methods

### 4.1. Bacteria, Nematodes and Cultivation Conditions

Appendix A list the bacterial strains and plasmids used in this work, and Appendix A lists the primers. *Escherichia coli* strains and *P. donghuensis* strains were grown in Luria-Bertani (LB) medium at 37 °C and 30 °C, respectively. When necessary, antibiotics were added at the following final concentrations: for *E. coli* strains, 50 μg/mL kanamycin and 10 μg/mL gentamicin; for *P. donghuensis* strains, 25 μg/mL chloramphenicol, 50 μg/mL gentamicin and 50 μg/mL kanamycin.

All *C. elegans* strains were purchased from the Caenorhabditis Genetics Center (CGC, University of Minnesota, Twin Cities, USA) and maintained in standard conditions at 20 or 25 °C. The wild-type Bristol N2, ZD101 *tir-1*(qd4) III, KU4 *sek-1*(km4) X and KU25 *pmk-1*(km25) IV worms were obtained from the Caenorhabditis Genetic Center. *C. elegans* was sustained on nematode growth medium (NGM) agar plates with overnight cultures of *E. coli* OP50 [41] as the food source and then incubated for 8 h at 37 °C. To obtain synchronous day-1 adult worms, the eggs laid over half an hour were collected and grown at 22 °C. Worm stocks were subjected to bleach treatment to remove contaminants [43], and worms from the generation after bleaching were used for experiments.

### 4.2. Comparative Genomics Analysis

The whole genome sequences of seven strains were processed and analysed by BGI (Shenzhen, China), and CD-HIT software was used for clustered, which enables the rapid clustering of similar proteins, with a threshold of 50% pairwise identity and a length difference cutoff of less than 30% amino acids. CD-HIT stands for Cluster Database at High Identity with Tolerance, the program takes a fasta format sequence database as input and produces a set of non-redundant (nr) representative sequences as output. In addition, CD-HIT outputs a cluster file, documenting the sequence groupies for each nr sequence representative [44,45]. These seven strains were *P. donghuensis* HYS (GenBank: AJJP01000001.1), *P. aeruginosa* PA14 (GenBank: CP000438.1), *P. aeruginosa* PAO1 (GenBank: AE004091.2), *P. putida* KT2440 (GenBank: AE015451.2), *P. syringae* B728a (GenBank: CP000075.1), *P. syringae* pv. *tomato* DC3000 (GenBank: AE016853.1) and *P. fluorescens* Pfo-1 (GenBank: CP000094.2). Gene families were constructed by integrating multiple software: the protein sequences were aligned in BLAST, the redundancy was eliminated by Solar, and gene family clustering treatment for the alignment results was carried out with Hcluster sg software. The sequences were processed and analyzed by BGI (Shenzhen, China). In addition, genes were aligned against several databases, including the NCBI nonredundant protein database (http://www.ncbi.nlm.nih.gov (accessed on 16 August 2021)) and the KEGG pathway database (http://www.genome.jp/kegg (accessed on 16 August 2021)) by BLASTP.

### 4.3. Screening of Virulence Factors

The virulence factor database (VFDB, http://www.mgc.ac.cn/VFs/ (accessed on 16 August 2021)) is devoted to providing the scientific community with a comprehensive repository and online platform for deciphering bacterial pathogenesis. The virulence factor database (VFDB) is a comprehensive database integrating virulence factors of pathogenic bacteria. At present, it has the virulence gene sequence information of 30 genera (more than 100 kinds of medical pathogens). It involves 24 common pathogenic bacteria that are more important in medicine, including *Pseudomonas*, *Shigella, Escherichia*, *Salmonella*, and *Bacillus*. The prediction of virulence factors mainly depends on the specific software of the comparison algorithm; by using BLASTP against the VFDB, the comparison results can show the corresponding location of each gene annotated in the VFDB database and the description of virulence-related functions.

### 4.4. Construction of Mutant and Complement P. donghuensis HYS

Primers with restriction enzyme sites were designed to amplify fragments located upstream and downstream of each target gene (Appendix A). A bacterial genome DNA purification kit (Promega, Madison, WI, USA), gel extraction DNA purification kit (Omega Bio-Tek, Norcross, GA, USA), plasmid DNA extraction kit (Omega Bio-Tek, Norcross, GA, USA) After two amplified fragments were digested with primer-specific restriction enzymes (quick restriction digestion enzymes purchased from Thermo Fisher Scientific) (Waltham, MA, USA) and ligated into the suicide vector pEX18Gm, the correct recombinant plasmid was transformed from *E. coli* S17-1 (λpir) into *P. donghuensis* HYS via conjugation [46]. The target gene was knocked out, and selection for double recombinants was performed on LB agar plates containing 10% (wt/vol) sucrose. The correct gene deletion mutants were further confirmed by PCR and sequencing. Related complementation mutants were constructed by ligating the Shine-Dalgarno sequences and open reading frames (ORFs) of the target genes into pBBR1MCS-2 [47]. After conjugation with the corresponding mutants, correct monoclonal targets were selected with double-antibiotic treatments and stored at −80 °C.

### 4.5. Construction of Point Mutations in P. donghuensis HYS

Point mutations were introduced via homologous recombination using the suicide plasmid pEX18Gm. The amplified upstream and downstream fragments contained a homologous sequence of at least 20 bp, and the mutation site was in the homologous sequence (Appendix A). The flanking segments were then in-frame ligated by fusion PCR and inserted into the suicide plasmid pEX18Gm with the restriction enzyme. Then, the correct recombinant plasmid was transformed from *E. coli* S17-1 (λpir) into *P. donghuensis* HYS via conjugation [15], and the mutation site in the plasmid was inserted into the genome of *P. donghuensis* HYS. Finally, selection for double recombinants was performed on LB agar plates containing 10% (*w*/*v*) sucrose. The point mutants were further subject to PCR and sequencing to confirm they had the correct identity.

### 4.6. Slow-Killing Assays of C. elegans

In vivo infection assays and slow-killing assays of *C. elegans* were performed as previously described [15,48]. Approximately 100 synchronized adult worms were added to five plates and incubated at 22 °C. *C. elegans* worms were considered dead or alive based on their response to being touched with a platinum wire. Abnormal deaths (crawling up the sides of the plates and drying out) were noted, and living *C. elegans* were transferred to new NGM plates every day. Worms were scored daily under an SZM-45B1 stereomicroscope (Sunny Optical, Yuyao, China) [49]. Kaplan–Meier survival curves were generated using the software IBM SPSS version 18.0 (SPSS Inc., Chicago, USA). Survival rates were compared and represented as the LT_50_ value. HYS or mutants were cultured in LB at 30 °C for 12 h, and then transferred 200 μL bacterial culture (3 × 10^9^ CFU/mL) to per NGM plate as food for *C. elegans*. The plates were dried before use. HYS was used as a positive control, and *E. coli* OP50, a traditional food source for *C. elegans*, was used as a negative control. Each experiment was replicated three to six times.

### 4.7. Growth Curve Assay

In order to eliminate the possible effect of strain growth on virulence, the growth curves of the strains on the NGM plate were determined [50,51]. Freshly isolated strains were inoculated in nutrient-rich LB liquid medium at 30 °C and shaken and cultured for 12 h. The *OD*_600_ of the bacterial solution was determined and adjusted to be consistent, and 200 μL of the bacterial solution was added to the plate. Three replicates were set for each strain. The strains on the NGM plate were grown at 22 °C for 12 h, the concentration was appropriately diluted, and the growth curve was drawn.

### 4.8. Transmission Electron Microscopy

In order to reveal bacterial surface sugars, transmission electron microscopy was used at the highest possible resolution. Bacteria were washed three times and resuspended to an *OD*_600_ of 0.2. Bacteria were harvested and fixed in 3% (*w/v*) glutaraldehyde and 0.075% (*w/v*) ruthenium red in 0.1 M PBS for one hour in the dark, and they were post-fixed for two hours in the dark in 0.075% (*w/v*) ruthenium red and 1% (*w/v*) osmium tetroxide. Samples were dehydrated through graded ethanol, transferred to pure Spurr resin and blocked. Specimens were observed under transmission electron microscope (JEM-1400 plus, Japan Electronics Co. LTD, Shojima, Tokyo, Japan).

### 4.9. C. elegans Bacterial CFU Analysis

*C. elegans* were exposed to *P. donghuensis* HYS under the same conditions as used in the survival assay. Bacterial CFUs within the nematode gut were counted according to a method described in the literature [52]. After a defined period, 6 replicates of 10 worms each were transferred to M9 solution containing 25 mM levamisole to paralyze the worm and stop pharyngeal pumping. Worms were washed twice with an antibiotic solution containing chloramphenicol (25 μg/mL) in levamisole (25 mM), followed by 1 h of incubation in the antibiotic solution to kill bacteria present on the exterior of the worm. After three washes with a solution of 25 mM levamisole to remove the antibiotics, worms were lysed with a motorized pestle. Lysates were serially diluted in M9 solution and plated on Luria–Bertani plates containing chloramphenicol (25 μg/mL) to select for *P. donghuensis* HYS and select against OP50. After overnight incubation at 30 °C, colonies of *P. donghuensis* HYS were counted to determine CFU per worm.

### 4.10. Microscopy

Experiments involving worms were performed by utilizing the SZM-45B1 stereomicroscope (Sunny Optical, Yuyao, China).

### 4.11. Statistical Analysis

All data were presented as the means  ±  standard deviation (SD), and each experiment was performed at least three times independently. Survival curves were plotted in IBM SPSS version 18.0 (SPSS Inc., Chicago, USA) using the Kaplan–Meier method. Statistical analyses were performed using OriginPro 9.0 (OriginLab, USA). Significant differences were evaluated using the Student’s *t*-test, *p* < 0.05 was considered statistically significant.

### 4.12. Accession Numbers

The GenBank accession numbers for the genes *gtrA*, *gtrB*, and *gtrII* from *P. donghuensis* HYS are UW3_RS0104075, UW3_RS0104080, UW3_RS26470, respectively.

## Figures and Tables

**Figure 1 ijms-22-10741-f001:**
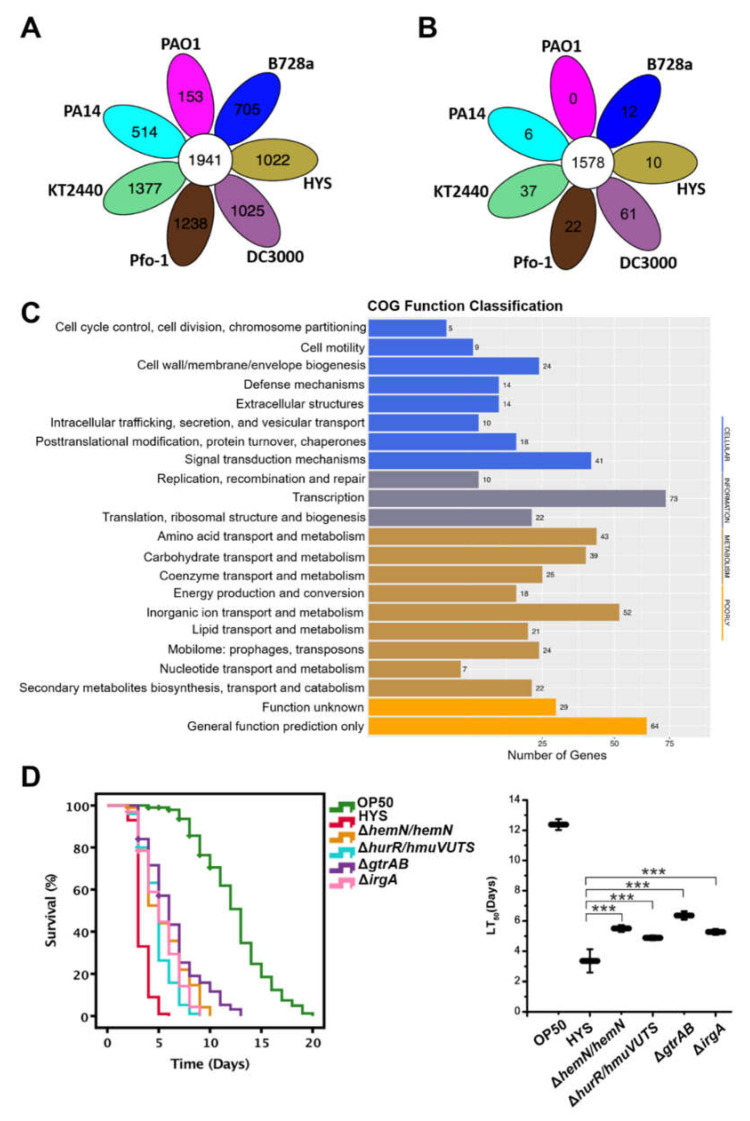
UpSet plots of pan gene set and the number of homologous gene families in each species. (**A**) UpSet plot of pan gene set among seven strains. (**B**) UpSet plot of the number of homologous gene families among seven strains. Each ellipse represents a sample, and the data in each region represent the number of clusters that only occur in samples in this region. A cluster represents a group of genes with sequence length differences of less than 30% and more than 50% similarity. (**C**) COG functional classification of unique genes in *P. donghuensis* HYS. (**D**) The functions of 42 unique virulence genes in *P. donghuensis* HYS were assessed using slow-killing experiments. Knockout strains with significant virulence reduction were listed, while other knockout strains without significant virulence reduction are listed in Appendix A. Approximately 100 synchronized adult worms were added to five plates for each bacterium. Survival curves were plotted in IBM SPSS version 18.0 using the Kaplan–Meier method. Data are presented as the mean  ±  standard deviation from three independent experiments. *** *p* < 0.001, Student’s *t*-test.

**Figure 2 ijms-22-10741-f002:**
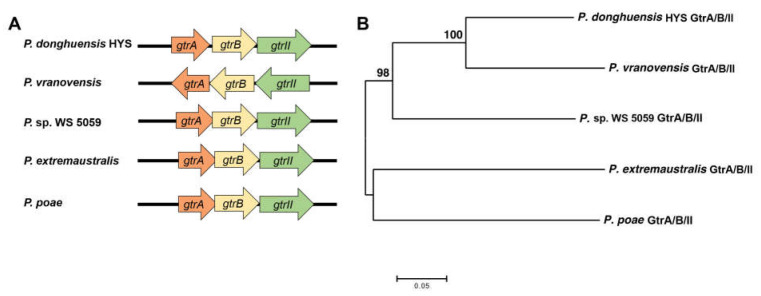
Arrangement and phylogenetic analysis of GtrA/B/II in *Pseudomonas*. (**A**) Arrangement of GtrA/B/II in five strains. (**B**) Phylogenetic analysis of *P. donghuensis* HYS GtrA/B/II using amino acid sequences with MEGA7.0 program.

**Figure 3 ijms-22-10741-f003:**
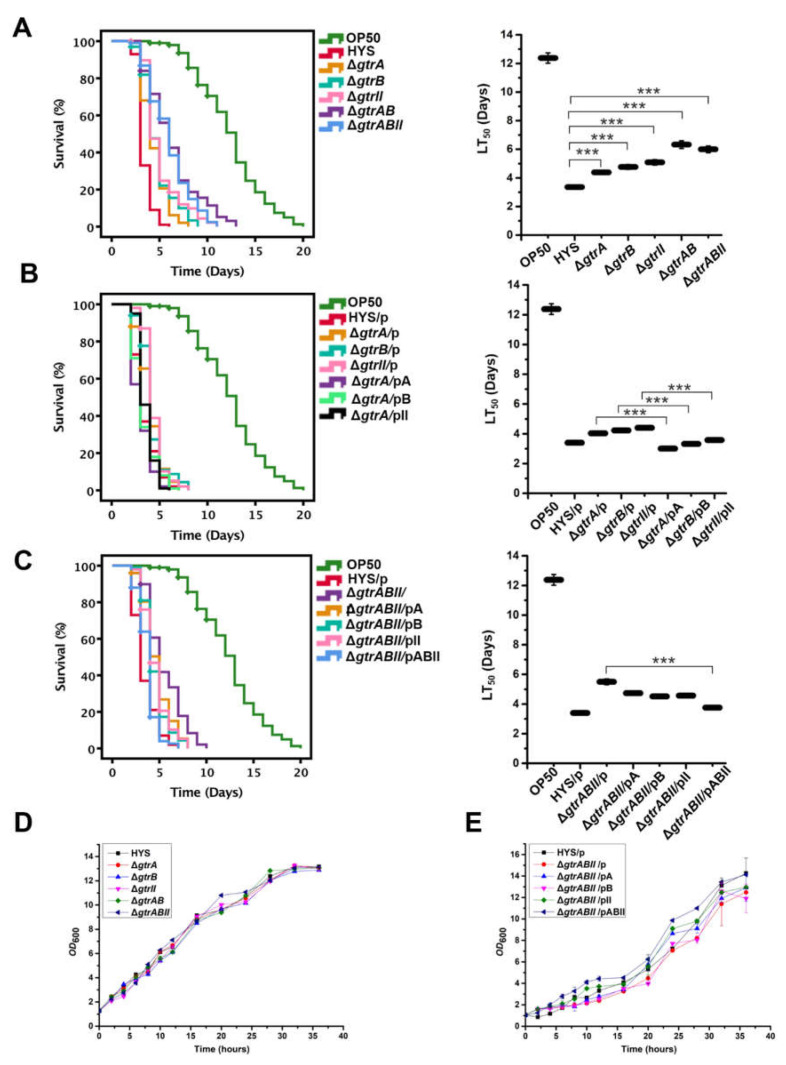
The *gtrA*/*B*/*II* gene cluster is associated with pathogenicity in the *C. elegans* slow-killing model. (**A**) Mutants of *gtrA*, *gtrB* and *gtrII* were tested using slow-killing experiments. (**B**) The functions of *gtrA*, *gtrB* and *gtrII* were further confirmed by using single-gene complementation in the single-knockout strains. p represents the expression plasmid pBBR1MCS-2; pA, pB and pII represent the recombinant plasmids pBBR2-*gtrA*, pBBR2-*gtrB* and pBBR2-*gtrII*, respectively. (**C**) We also performed single-gene complementation and three-gene complementation in the triple-knockout strains. pABII represents the recombinant plasmid pBBR2-*gtrABII*. Approximately 100 synchronized adult worms were added to five plates for each bacterium. Survival curves were plotted in IBM SPSS version 18.0 using the Kaplan–Meier method. (**D**,**E**) Growth curves of *P. donghuensis* HYS and mutants. Data are presented as the mean  ±  standard deviation from three independent experiments. *** *p* < 0.001, Student’s *t*-test.

**Figure 4 ijms-22-10741-f004:**
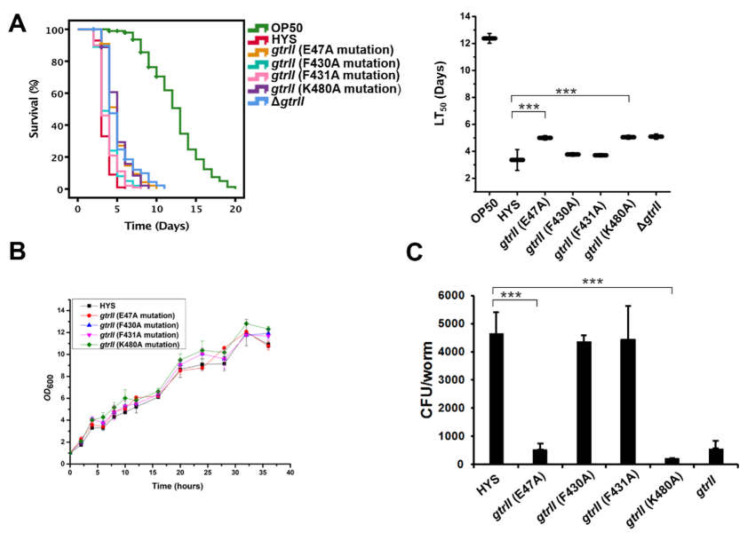
Critical GtrII residues control the virulence function of GtrII in *P. donghuensis* HYS. (**A**) The effects of E47A, F430A, F431A and K480A point mutations in *P. donghuensis* HYS were tested using slow-killing experiments. Approximately 100 synchronized adult worms were added to five plates for each bacterium. Survival curves were plotted in IBM SPSS version 18.0 using the Kaplan–Meier method. (**B**) Growth curves were also constructed in this killing experiment. (**C**) *P. donghuensis* HYS CFU in the gut of nematodes exposed to point mutation strains. Data are presented as the mean  ±  standard deviation from three independent experiments. *** *p* < 0.001, Student’s *t*-test.

**Figure 5 ijms-22-10741-f005:**
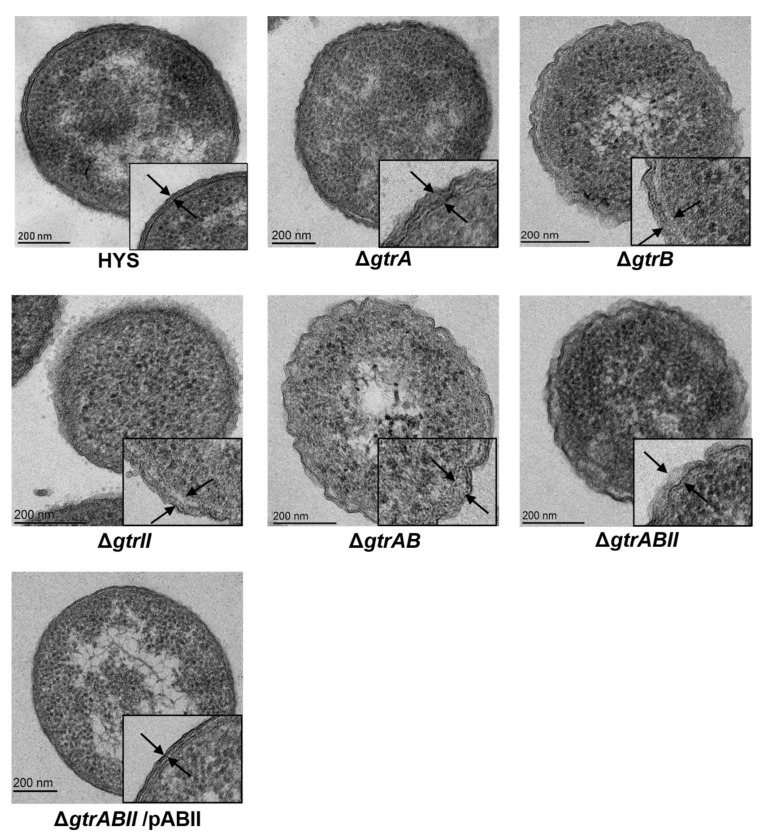
Surface topology of strains revealed by transmission electron microscopy after treatment with ruthenium red. The electron-dense material on the bacterial surface is indicated between the arrows. In Δ*gtrA*, Δ*gtrB*, Δ*gtrII*, Δ*gtrAB* and Δ*gtrABII*, the surface material is less compact than HYS.

**Figure 6 ijms-22-10741-f006:**
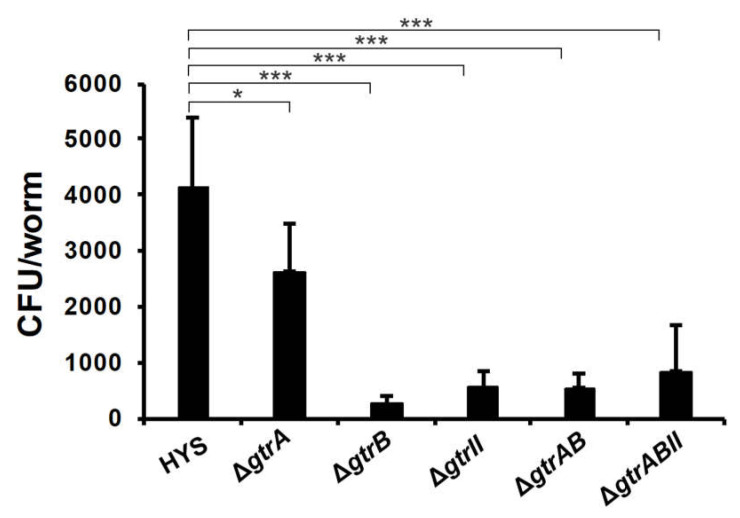
*P. donghuensis* HYS CFU in the gut of nematodes exposed to deletion mutants. Each symbol represents the average of 6 biological replicates of 10 worms. A comparable initial inoculum was ensured for the exposure of wild-type worms to HYS, Δ*gtrA*, Δ*gtrB*, Δ*gtrII*, Δ*gtrAB* and Δ*gtrABII* for 24 h. Data are presented as the mean  ±  standard deviation from three independent experiments. * *p* < 0.05 and *** *p* < 0.001, Student’s *t*-test.

**Figure 7 ijms-22-10741-f007:**
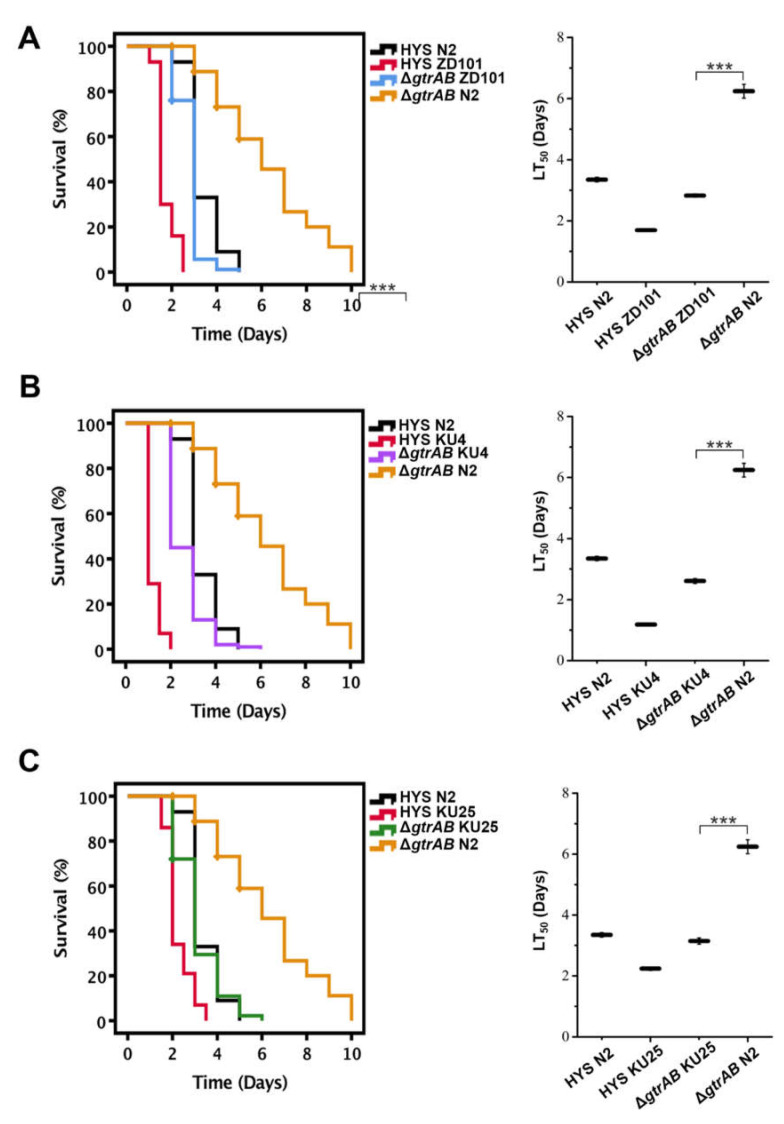
Correlation between GtrA/B/II-involved virulence and worm innate immunity. (**A**) N2 and mutant worms deficient in the Tir-1 pathway, (**B**) the SEK-1 pathway and (**C**) the PMK-1 pathway of the p38 MAPK pathway were fed *P. donghuensis* HYS or *gtrAB*. Approximately 100 synchronized adult worms were added to five plates for each bacterium. Survival curves were plotted in IBM SPSS version 18.0 using the Kaplan–Meier method. Data are presented as the mean  ±  standard deviation from three independent experiments. *** *p* < 0.001, Student’s *t*-test.

**Table 1 ijms-22-10741-t001:** Information on strains with the same arrangement of GtrA/B/II as *P. donghuensis* HYS.

Species	Version	Isolation Source	Isolation Country	Reference
*P. donghuensis* HYS	NZ_AJJP00000000.1	Lake	China	Gao et al., 2012 [12]
*P. vranovensis*	NZ_MOAM00000000.1	River	USA	Tao et al., 2016 [30]
*Pseudomonas* sp. WS 5059	NZ_JAAQWO010000002.1	Raw milk	Germany	Maier et al., 2020 [31]
*P. poae*	NZ_PCQN01000008.1	Rotting apple	Germany	Schulenburg et al., 2017 [32]
*P. extremaustralis*	NZ_AHIP01000006.1	Temporary water pond in Antarctica	Argentina	Tribelli et al., 2012 [33]

## Data Availability

All data generated or analysed during this study are included in this published article (and its Appendix A).

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
