# Peer review of "Pseudomonas donghuensis HYS gtrA/B/II Gene Cluster Contributes to Its Pathogenicity toward Caenorhabditis elegans"

_ijms, 2021, doi:10.3390/ijms221910741_

Round 1

Reviewer 1 Report

Abstract

Line 8-9: Please revise this sentence “Pseudomonas donghuensis HYS, which has more virulent than P. aeruginosa towards Caeno- 8 rhabditis elegans and has an unclear virulence mechanism”

Line 26: replace word “special” with unique

Introduction

Line 32-36: Please revise this paragraph with appropriate words. Replace words “special”, “bad” “dealing”, “attacks’… with appropriate words

Line 37: Revise the sentence, replace word “kind” with appropriate word

Line 45: Correct the sentence

Line 48-54: Revise this paragraph for better understanding to readers

English needs to be improved throughout this section

Results & Methods

Line 78-81: Please cite the earlier study

How many (n=?) C. elegans worms were used for slow-killing assays? How many repeats were performed, please clarify

Do authors did the statistical analysis of survival curves using log-rank test?

Why did authors used E. coli OP50 as a control instead of P. aeruginosa strain? No data was presented or cited comparing P. donghuensis HYS virulence as compared to other Pseudomonas strains.

What is the rationale behind choosing mutant worms ZD101, KU4 258 and KU25? please clarify

Cite the reference for CD-HIT cluster analysis, provide the rationale and additional details regarding this analysis

Line 99: Did authors knocked out all 42 specific virulence genes and performed slow-killing assays? Please clarify

What was the bacterial inoculum used in slow-killing assay? Please clarify

Details of the methods would be better throughout methods section

Discussion

Line 280-281: Please cite the correct reference

Extensive discussion of the observed results is needed

Reviewer 2 Report

This manuscript describes the identification of a virulence operon in Pseudomonas donghuensis, a mesophilic bacterium that was originally isolated from lake water. The gtrABII operon was found in P. donghuensis and 4 other Pseudomonas species. A similar operon has been found in Shigella flexneri and is involved in the modification of the O-antigen in LPS. The authors have generated deletion mutants of individual genes and combinations therof. These were tested in a C. elegans worm infection model and showed decreased virulence. Immune-deficiency mutants of C. elegans were also tested.

The results are novel and interesting, although somehow derivative as this gene cluster has been analysed in Shigella before. However, I have some concerns about the rigour of the methodology.

  1. It is not clear how many worms per group were used and how many independent experiments were done. The text states “n=3’. Does this mean 3 repeat experiments with 10 worms per group? Is the Student’s test the correct statistical analysis?
  2. Figure 1A and B: these are not really Venn-diagrams. No overlaps.
  3. Figure 1C: how were these genes characterized if they are unique to this species?
  4. Figure 1D: in the legend it says that 42 genes were analysed, but the figure only shows 4. What is the significance of this result? Apart from the gtrAB deletion, nothing is further described in the manuscript. Was there a negative control (e.g. PBS)?. What do the error bars represent (average or median)? What are the values for ***?
  5. Figure 2B: what are the numbers (98, 100)?
  6. It is not clear to me how the GtrII protein residues were selected for alanine conversion. How is it known what the critical residues are if its function is unclear?
  7. The TEM analysis does not convincingly show that the LPS O-antigen is shorter in the mutants. The authors should try to purify LPS from the strains and apply mass-spectroscopy.
  8. Page 9, line 226: how do the authors conclude that T3SS is exposed on the mutant strains? Also in discussion on page 13. There is no experimental evidence.
  9. Figure 7, right panel: what does the Y-axis show? Are the differences between N2 and worm mutants statistically significant for each of the Pseudomonas mutants?
  10. Discussion: the authors should describe the 3 immune-deficiency mutants and why they believe that virulence mechanism of the Gtr system might be connected to innate immune responses and how.
  11. How were the alanine conversion mutants generated?
  12. elegans that died abnormally (bagged or crawled up the sides of the plates and dried out) were noted.” What does this mean? Excluded? How often did that happen? The authors need to clarify how many worms were actually used for each experiment.
  13. “Because the function of gtrII is not clear, gtrII is not screened in VFDB data base and COG database.” Please explain.
  14. The introduction contains some very unusual terms and statements which require clarification: “virulence is a special survival strategy ….”, Pseudomonas is a kind of obligate …”, “HYS has a competitive system to absorb ….”, “ may contain toxic molecular gene clusters …”,

Round 2

Reviewer 1 Report

Line 32-33: Please redefine the term "Virulence", it is not a survival strategy. 

Line 208: Please specify how much is 200 μl bacterial culture in terms of CFU/mL.

Reviewer 2 Report

This is an improved version of the manuscript. There are still a few issues that need to be addressed.

Introduction

“Virulence is a survival strategy for pathogens under harsh conditions”. That is not really the definition for virulence. Virulence is a pathogen’s ability to cause harm/disease.

It [C. elegans] is amenable to high-throughput screening, and its genome shares high homology with human disease-related genes [17, 18, 19, 20]. I don’t think that this statement is correct. Insects have become interesting infection models due to several advantages and one of them is the presence of a functional innate immune response. This doesn’t mean that there are any genes with high DNA sequence homologies.

Fig 1D: please clarify in the figure legend that only some results for the 42 mutant strains are shown and justify why. Please add the information of the number of worms for each experiment and how the survival curves were plotted. This information has been added to the main text but should also appear in the figure legends. This also applies to other figures.

Q3. Figure 1C: how were these genes characterized if they are unique to this species?

Our reply: Thanks for your comments. The characteristics of these genes are as follows: Through comparative genomic analysis, these unique genes exist only in the genome of HYS, while the genomes of the other six model Pseudomonas do not contain these genes. Apologies, my question might have been confusing. What I don’t understand is, if these genes are unique to HYS and not found in any other species, how can you classify them into functional groups?

Q6. It is not clear to me how the GtrII protein residues were selected for alanine conversion. How is it known what the critical residues are if its function is unclear? Our reply: Thanks for your comments. We refer to the method of this reference. “In Shigella flexneri, three critical residues (Glu 40, Phe 414, and Lys 478) are conserved in Gtr[type]. Each residue was converted into alanine, which destroyed GtrII-mediated O antigen modification” [30]. However, in their response to Q13, the authors wrote: However, due to the low sequence homology and protein length difference between GtrII of HYS and Shigella flexneri, GtrII has not been annotated in VFDB, KEGG, COG or GO databases through analyses genome sequencing data. How can functionally important residues be selected when there is very limited homology between the two proteins? How similar are the proteins between the two species?  The authors should provide a sequence alignment of the two proteins and highlight the selected residues.

There are still some issues around the interpretation of the TEM analysis. I don’t think the results are conclusive enough to indicate shorter LPS. For example, the surface of ΔgtrABII/pABII, which displays enhanced glucosylation …”. There is no evidence showing enhanced glycosylation. “In wild-type HYS with truncated LPS, which might enhance bacterial invasion. The extended LPS in the absence of glycosylation impaired its invasive capacity and reduced virulence.” Invasion of host cells via T3TT is an important virulence mechanism in Shigella, but is this also true for Pseudomonas donghuensis? P. aeruginosa has been shown to invade cells but this appears to contribute to chronic disease (UTI) rather than an acute symptom that could be measured in the C. elegans model. In fact, Pseudomonas strains with longer LPS chains were found to be more virulent due to interference with complement deposition, in particular the MAC complex. The authors need to either provide more conclusive evidence that gtr is responsible for LPS modification or tone down their discussion.

The term ‘toxic’ or toxicity’ is mentioned several times throughout the manuscript but I don’t believe this is correct in this context as there are no toxin-mediated effects reported. “Virulence” might be a better term.

Round 3

Reviewer 2 Report

No further comments